# Establishment and optimization of an *in vitro* guinea pig oocyte maturation system

Minhua Yao[1], Zhaoqing Gong[1], Weizhen Xu[1], Xinlei Shi[1], Xiaocui Liu[1], Yangyang Tang[1], Siyu Xuan[1], Yanping Su[1], Xinghua Xu[1]*, Mingjiu Luo[2]*, Hongshu Sui[1]*

**1** Department of Histology and Embryology, School of Basic Medicine, Shandong First Medical University & Shandong Academy of Medical Science, Jinan, Shandong, P. R. China, **2** Shandong Provincial Key Laboratory of Animal Biotechnology and Disease Control and Prevention, College of Animal Science and Veterinary Medicine, Shandong Agricultural University, Tai'an City, P. R. China

☯ These authors contributed equally to this work.
\* xhxu@sdfmu.edu.cn (XX); luomj@sdau.edu.cn (ML); hssui@sdfmu.edu.cn (HS)

**Data Availability Statement:** All relevant data are within the paper and its Supporting Information files.

**Funding:** This work was supported by the National Natural Science Foundation of China grants

## Abstract

Guinea pigs are a valuable animal model for studying various diseases, including reproductive diseases. However, techniques for generating embryos via embryo engineering in guinea pigs are limited; for instance, *in vitro* maturation (IVM) technique is preliminary for guinea pig oocytes. In this study, we aimed to establish and optimize an IVM method for guinea pig oocytes by investigating various factors, such as superovulation induced by different hormones, culture supplementation (e.g., amino acids, hormone, and inhibitors), culture conditions (e.g., oocyte type, culture medium type, and treatment time), and *in vivo* hCG stimulation. We found that oocytes collected from guinea pigs with superovulation induced by hMG have a higher IVM rate compared to those collected from natural cycling individuals. Moreover, we found that addition of L-cysteine, cystine, and ROS in the culture medium can increase the IVM rate. In addition, we demonstrated that *in vivo* stimulation with hCG for 3–8 h can further increase the IVM rate. As a result, the overall IVM rate of guinea pig oocytes under our optimized conditions can reach ~69%, and the mature oocytes have high GSH levels and normal morphology. In summary, we established an effective IVM method for guinea pig oocytes by optimizing various factors and conditions, which provides a basis for embryo engineering using guinea pigs as a model.

## Introduction

Guinea pigs are an ideal animal model for studying a variety of diseases related to immunology, pharmacology, nutritional science, toxicology, and the respiratory system [1]. They also show promise as a model for reproductive diseases, despite being less popular than other rodents like mice and rats. This is because guinea pigs have a longer estrus cycle of 16–18 days and a longer gestation duration of approximately 65 days [2, 3]. Notably, however, guinea pigs exhibit follicular and luteal phases that are similar to those of humans and other large mammals, such as pigs, cattle, and sheep, throughout the complete estrous cycle [4]. For example,

#32072738 (ML) and #81670004 (HS), the Shandong Medical and Health Science and Technology Development Project #202001010899, and the Doctoral Startup Fund of Shandong First Medical University #001003053 (XX).

**Competing interests:** The authors have declared that no competing interests exist.

in the guinea pig estrous cycle, ovarian follicles will experience two stages of follicular wave peaks at 10–11 days and 16 days, respectively [5, 6]. Moreover, guinea pig blastocysts undergo interstitial implantation similar to human blastocysts, forming syncytiotrophoblasts [7, 8]. These characteristics make guinea pigs suitable for modeling and studying diseases in large mammals.

The requirement for inducing *in vitro* maturation (IVM) of oocytes is not very demanding; it does not require the oocytes to be at a specific sexual maturity state. Moreover, oocytes can be readily obtained from the antral follicles of the ovary [9]. Therefore, IVM of oocytes has advantages over *in vivo* maturation stimulated using hormones. High quality and high efficiency of IVM from guinea pig oocytes are essential for *in vitro* fertilization (IVF), embryonic development, and nuclear transfer cloning in guinea pigs. However, compared with other mammals like mice [10], cattle [11], and sheep [12], the success rate of IVF in guinea pigs is relatively low [4]. Furthermore, few studies have explored IVM of guinea pig oocytes, possibly because of the poor productive capacity of guinea pigs that can hinder the maturity of oocyte nuclei and cytoplasm [8]. Although some studies have managed to carry out IVM and IVF in guinea pigs, the quality of the developed embryos was lower than the quality of those developed from natural maturation due to some unknown reasons [13, 14].

Previous research has shown that many factors, such as superovulation induced by certain hormones, culture supplementation (e.g., amino acids, hormones, and inhibitors) and conditions (e.g., oocyte type, medium type, and culture duration), and *in vivo* stimulation with hormones can affect IVM rate. For example, Martín-Coello *et al*. showed that superovulated oocytes have an overall higher IVM rate compared to oocytes collected from natural cycling animals [15]. Hormones are critical for the maturation of oocytes and thus are usually added to the culture medium for IVM [16]. Previous studies have also shown that culture conditions, such as supplementation of certain amino acids (e.g., L-cysteine, cystine), as well as different culture durations, can affect the IVM rate in cows [17] and mice [18]. Certain inhibitors, such as roscovitine (ROS) and 3-isobutyl-1-methylxanthine (IBMX), can also affect the IVM rate of oocytes in cattle [19]. ROS is a cyclin-dependent kinase inhibitor, while IBMX is a nonspecific inhibitor of cAMP and cGMP phosphodiesterases; both inhibitors can be used to maintain meiotic arrest of oocytes and thus mimic the environment that inhibits the follicles and keep the oocyte in the germinal vesicle (GV) stage, improving their embryonic developmental potential and finally increasing the IVM rate [20]. Moreover, some studies also showed that oocytes that are induced *in vivo* for certain durations (e.g., hCG) have an overall higher IVM rate [21]. Thus, in this study, we aimed to explore how these factors can enhance the IVM rate of guinea pig oocytes, thus providing a basis for further research on diseases such as reproductive diseases using guinea pig embryos.

## Materials and methods

### Culture medium, reagents, chemicals, and equipment

The media, chemicals, inhibitors and reagents used were as follows: M199 medium (#M0393, Sigma, USA), M2 medium, HEPES-buffered CZB medium (HCZB) (#H4034, Sigma, USA), FBS (#10099141C, Gibco, USA), PBS, ROS (#557354, Sigma, USA), IBMX (#I5879, Sigma, USA), and pregnant mare's serum gonadotropin (PSMG) (#110254564, ShuSheng Inc, China), follicle-stimulating hormone (FSH) (#H10940097, Livzon Pharmaceutical Group Inc, China), luteinizing hormone (LH) (#(2016)110254634, ShuSheng Inc, China), penicillin (#H37020079, Shandong Lukang Pharmaceutical Co., Ltd., China), human chorionic gonadotrophin (hCG) (#110251282, NSHF Inc, China), human menopausal gonadotrophin (hMG) (#H44020668, Livzon Pharmaceutical Group Inc, China), chloral hydrate (Q/12HB 4218–2017, Kermel,

China), streptomycin (#H37020187, Shandong Lukang Pharmaceutical Co., Ltd., China), pyruvic acid (#W297003, Sigma, USA), L-cysteine (dissolved in HCL) (#G7602, Sigma, USA), cystine (dissolved in HCL) (#C7602, Sigma, USA), polyvinyl alcohol (PVA) (#P8136, Sigma, USA), paraformaldehyde (PFA) (#A8020, Solarbio, USA), Triton X-100 (#T8200, Solarbio, USA), BSA (#A1933, Sigma, USA), Hoechst 33342 (Y35467, Shanghai yuanye Bio-Technology Co., Ltd, China), Cell Truck Blue (CMF2HC; 4-chloromethyl-6,8-difluoro-7-hydroxycoumarin) (#C12881, Life Technology, USA). FITC-labeled anti-α-tubulin antibody (#L7381, Sigma, USA). The equipment used was as follows: cell culture incubator (NU-4750, NUAIRE, USA), inverted fluorescence microscope (TS2, Nikon, Japan), and laser confocal microscope (A1R MP, Nikon, Japan).

## Guinea pigs

The study was approved by the Animal Care and Use Committee of Shandong First Medical University (Approval No. W202111220327). Female guinea pigs aged around 20 days, which weighed around 80 g and were not sexually matured yet, were purchased from Jinan Pengyue Experimental Animal Breeding Co., Ltd. (Jinan, China). The guinea pigs were housed in a temperature-regulated (22–23˚C) animal facility that followed a 12:12 h dark:light cycle, with a relative humidity of 65%. The guinea pigs were given free access to commercial food (pellet form) enriched with vitamin C and tap water containing 0.2 mg/ml of vitamin C.

## Superovulation in guinea pigs

Superovulation in 20-day-old female guinea pigs was induced by subcutaneous injection of different hormones. For different experimental conditions, either 500 μL of FSH (5 IU per kg weight in PBS) (hereafter as 5 IU/kg), 500 μL of PMSG (5 IU/kg in PBS), or 500 μL of hMG (5 IU/kg in PBS) was subcutaneously injected into the back of the guinea pigs' neck for 1 day, or 500 μL of hMG (5 IU/kg in PBS) was subcutaneously injected in the same area for 2 or 3 consecutive days. Cumulus oocyte complexes (COCs) containing oocytes were dissected at specific time points and used for subsequent IVM experiments.

## Collection of COCs from natural cycling and superovulated guinea pigs

PBS and M2 medium were pre-warmed to 30˚C. Natural cycling or superovulated guinea pigs were anesthetized with 5% chloral hydrate at a dose of 0.2 ml per 10 g body weight, and their ovaries were dissected, collected, and briefly washed in pre-warmed PBS. After trimming, the ovaries were placed in pre-warmed M2 media. Then, follicles within the ovaries were punctured using a glass needle under a stereomicroscope, allowing the COCs containing oocytes to flow into the M2 media. Representative COCs are shown in Fig 1.

## COCs culturing and *in vitro* maturation

The collected COCs were first washed in the pre-warmed M2 medium and then cultured in either basic medium (i.e., M199 medium supplemented with 10% fetal bovine serum (FBS), 100 unit/ml penicillin, 50 μg/ml streptomycin, and 5.2 ng/mL of pyruvic acid) or basic medium supplemented with different components according to different experimental conditions. L-cysteine and cystine were prepared in concentrated stock solutions (200 μM and 100 μM, respectively) and diluted in the culture medium when used. To compare how different COC types affect the IVM rate, the basic medium was used (Table 1). To determine how different hormones in the culture medium affect the IVM rate, the basic medium supplemented with either PMSG (5 IU/mL), PMSG (5 IU/mL) + hCG (1 IU/mL), PMSG (5 IU/mL)

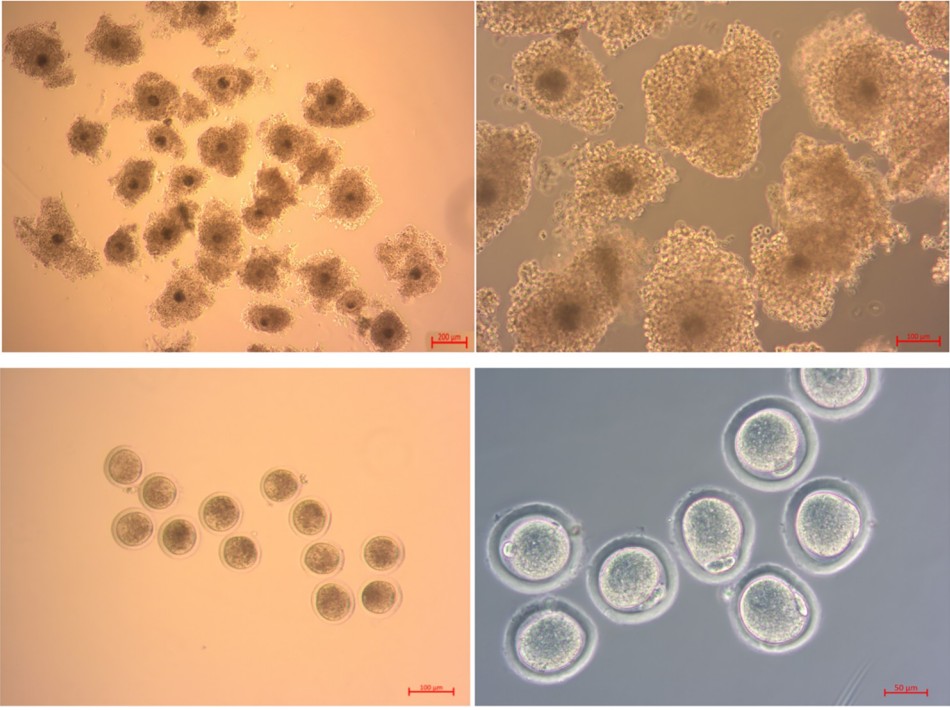

**Fig 1. Types of guinea pig oocytes and their meiotic progression during IVM.** (A,B,C) Compact COCs collected from the ovaries of guinea pigs. Type A oocytes have three or more layers of cumulus cells (A). Type B oocytes have 1–3 layers of cumulus cells (B). Type C oocytes have few or no cumulus cells (i.e., naked oocytes) (C). (D) Oocytes at the MII stage (matured oocytes) induced from Type A oocytes showing the first polar body (arrow). Scale bar = 200 μm in (A); 100 μm in (B, C); 50 μm in (D).

+ hCG (1 IU/mL) + FSH (1 IU/mL), or hCG (1 IU/mL) + FSH (1 IU/mL) + LH (5 IU/mL) were used (Table 2). To determine how different hormone-induced superovulation affect the IVM rate, the basic medium + hCG (1 IU/mL) + FSH (1 IU/mL) + LH (5 IU/mL) were used. To determine how different culture durations affect the IVM rate, the basic medium supplemented with hCG (1 IU/mL) + FSH (1 IU/mL) + LH (5 IU/mL) was used to culture the oocytes for 24 h, 36 h, and 48 h (Table 4). To determine how different amino acids and inhibitors affect the IVM rate, the basic medium without any supplementation or supplemented with either hCG (1 IU/mL) + FSH (1 IU/mL) + LH (5 IU/mL), hCG (1 IU/mL) + FSH (1 IU/mL) + LH (5 IU/mL) + L-cysteine (L-Cys hereafter) ((200 μM), hCG (1 IU/mL) + FSH (1 IU/mL) + LH (5 IU/mL) + L-Cys ((200 μM) + cystine (Cys hereafter) (100 μM), hCG (1 IU/mL) + FSH (1 IU/mL) + LH (5 IU/mL) + L-Cys ((200 μM) + Cys (100 μM) + IBMX (50 μM), or hCG (1 IU/mL)

**Table 1. Effects of different cumulus cell types (layers) on *in vitro* maturation of oocytes collected from natural cycling guinea pigs.**

| Cumulus type | Number of animals | Number of ocytes | GV (%) | pMI (%) | MI (%) | AI/TI (%) | MII (%) | Lysed (%) |
|---|---|---|---|---|---|---|---|---|
| A | 10 | 152 | 7.95±1.68[a] | 40.89±2.84[a] | 10.81±2.49[a] | 1.96±1.01[ab] | 11.57±2.48[a] | 26.82±2.97[a] |
| B | 10 | 132 | 7.72±2.71[a] | 40.58±4.71[a] | 19.38±2.68[a] | 0.00±0.00[a] | 8.33±2.26[ab] | 23.99±2.56[a] |
| C | 10 | 225 | 7.33±2.89[a] | 30.99±5.34[a] | 14.14±3.75[a] | 4.61±1.55[b] | 1.69±0.89[b] | 42.49±5.42[b] |

* Different superscript letters (a, b or ab) in the same column indicate statistical significance between different variables (P < 0.05). Type A oocytes have more than three layers of cumulus cells; type B oocytes have 1–3 layers of cumulus cells; type C oocytes have few or almost no cumulus cells. GV: germinal vesicle; pMI: pro-metaphase I; MI: metaphase I; AI: anaphase I; TI: telophase I; MII: metaphase II.

**Table 2. Effects of PMSG, hCG, FSH, and LH in the culture medium on the IVM rate of oocytes from natural cycling guinea pigs.**

| Culture medium | Number of animals | Number of oocytes | GV (%) | pMI (%) | MI (%) | A/T (%) | MII (%) | Lysed (%) |
|---|---|---|---|---|---|---|---|---|
| Basic medium | 10 | 112 | 7.95±1.68[a] | 40.89±2.84[a] | 10.81±2.49[a] | 1.96±1.00[ab] | 11.57±2.85[a] | 26.48±3.4[a] |
| Basic medium+PMSG | 6 | 59 | 6.70±1.26[a] | 40.99±2.32[a] | 20.68±3.25[b] | 1.51±1.51[ab] | 13.48±0.52[a] | 16.64±3.59[ab] |
| Basic medium+PMSG+HCG | 6 | 69 | 10.55±1.46[a] | 16.67±1.92[b] | 38.33±2.54[c] | 0.00±0.00[a] | 17.78±1.11[ab] | 16.67±1.92[ab] |
| Basic medium+PMSG+hCG+FSH | 6 | 70 | 5.86±1.64[a] | 14.32±2.88[b] | 44.26±0.75[c] | 5.57±0.92[ab] | 22.91±0.46[b] | 7.08±1.25[b] |
| Basic medium+hCG+FSH+LH | 6 | 60 | 3.82±1.90[a] | 16.78±0.47[b] | 39.39±2.52[c] | 4.62±2.39[b] | 25.10±1.34[b] | 10.29±1.16[b] |

* Different superscript letters (a, b or ab) in the same column indicate statistical significance between different variables (P < 0.05). IVM rates of type A oocytes cultured under different conditions were shown. FBS, fetal bovine serum; PMSG, pregnant horse serum gonadotropin; hCG, human chorionic gonadotropin; FSH, follicle-stimulating hormone; LH, luteinizing hormone; GV: germinal vesicle; pMI: pro-metaphase I; MI: metaphase I; AI: anaphase I; TI: telophase I; MII: metaphase II.

+ FSH (1 IU/mL) + LH (5 IU/mL) + L-Cys ((200 μM) + Cys (100 μM) + ROS (20 μM) was used (Table 5). ROS and IBMX were removed by replacing the old medium with fresh medium at different time points (i.e., 12 h and 24 h for ROS, and 6 h and 8 h for IBMX); after replacement of medium, the oocytes were further cultured for 6 h or 8 h after ROS removal and 12 h or 24 h after IBMX removal. To determine how *in vivo* injection of hCG (i.e., treatment duration and dose) affect the IVM rate, one day after induction of superovulation (i.e., injection of hMG for 3 consecutive days), either 5 IU/kg hCG (Table 6), or 0, 5 IU/kg or 10 IU/kg hCG (Table 7) was subcutaneously injected at the back of the neck; the COCs were then collected at the indicated timepoints (Table 6) or 8 h later (Table 7), and then cultured in basic medium hCG (1 IU/mL) + FSH (1 IU/mL) + LH (5 IU/mL) + L-Cys ((200 μM) + Cys (100 μM) + ROS (20 μM). To determine how different culture media affect the GSH levels in mature oocytes, the basic medium without any supplementation, or supplemented with hCG (1 IU/mL) + FSH (1 IU/mL) + LH (5 IU/mL), hCG (1 IU/mL) + FSH (1 IU/mL) + LH (5 IU/mL) + L-Cys ((200 μM) + Cys (100 μM), or hCG (1 IU/mL) + FSH (1 IU/mL) + LH (5 IU/mL) + L-Cys ((200 μM) + Cys (100 μM) + ROS (20 μM) were used. During culturing of oocytes, the medium was covered with paraffin oil, and incubated at 38.5˚C, 5% CO2, and 100% humidity. After a certain period of culturing, the matured oocytes were retrieved for further functional tests.

### *In vivo* stimulation with hCG

For in vivo stimulation of oocytes, 500 μL of hCG (5 IU/kg) diluted in PBS was subcutaneously injected into the back of the guinea pigs' neck one day following the superovulation procedure. Cumulus oocyte complexes (COCs) containing oocytes were then dissected at specific time points and used for subsequent IVM experiments.

### Examination of oocyte maturation status during IVM

Twenty-four hours after the IVM process was completed, the COCs were exposed to 0.1% hyaluronidase in D-PBS for 5 min. Then, the cumulus cells in the COCS were removed by washing with M199 medium supplemented with 10% FBS. The oocytes were stained with Hochest 33342, sealed with coverslips, and observed under a fluorescence microscope to determine their stages, i.e., GV, pMI, MI, A/T, or MII. Of note, MII, which is the indicator of IVM, was defined by the extrusion of the first polar bodies.

### Detection of glutathione (GSH) content in the oocyte cytoplasm

GSH is an important biomarker of oocyte quality and viability. To assess the impact of different culture media on GSH levels, oocytes cultured with different media were stained with Cell

Truck Blue (CMF2HC; 4-chloromethyl-6,8-difluoro-7-hydroxycoumarin; Life Technology) to determine cytoplasmic GSH levels. Briefly, in each treatment group, 10 oocytes were washed with PBS containing 0.1% PVA and then incubated with 100 μl of 10 μM Cell Truck Blue in the incubator at 37˚C, 5% CO2, and 100% humidity for 30 min. The oocytes were then washed with PBS containing 0.1% PVA three times for 5 min each, and observed under an inverted fluorescence microscope. The images were recorded and later analyzed using the Image J software (Version 1.45s; National Institutes of Health). The fluorescence intensities of the 10 embryos were used to evaluat GSH levels.

## Detection of microtubule morphology of mature oocytes

The oocytes were fixed in 4% paraformaldehyde solution, which was diluted in PBS, for 40 minutes at room temperature. Following this, the fixed oocytes were washed with PBS three times, each time for 5 min each time, and then treated with 0.2% Triton X-100/PBS for 20 minutes. After washing with PBS three times for 5 min each time, the oocytes were blocked in 2% BSA/PBS for 2 h, and then incubated with FITC-labeled anti-α-tubulin antibody (1:50) diluted in 2% BSA/PBS in a dark environment for 3 hours at room temperature or overnight at 4˚C. Next, the oocytes were washed with PBS three times for 5 minutes each time, and then stained with 10 μg/ml Hoechst 33342, which was diluted in PBS, for 5 minutes. After being washed with HCZB three times for 5 min each time, the oocytes were sealed with a coverslip, and observed and recorded under a laser confocal microscope (TCSSP2, Leica Laser Scanning Confocal System). The excitation wavelengths of Hoechst33342 and FITC were 405 nm and 408 nm, respectively.

## Statistical analysis

All data, including the number of oocytes and ratios of oocytes in the GV, pMI, MI, AI/TI, and MII stages, and degeneration rates of oocytes, were analyzed using the SPSS 20.0 (Chicago, USA). Percentage data in each replicate were arcsine transformed to mean ± standard error of the mean (SEM) and subjected to one-way ANOVA. Fluorescence intensity in treatment and control groups was compared using t-test, with $P < 0.05$ denoting statistical significance.

## Results

### Effects of COC type on the IVM rate of oocytes collected from natural cycling guinea pigs

Antral follicles were collected from the ovaries of 30 natural cycling guinea pigs. The COCs isolated from the follicles were divided into 3 types, i.e., A, B, and C (Fig 1), depending on the number of layers of cumulus cells. The COCs, which refer to collected oocytes with their cumulus cells, were cultured in the basic medium. The resulting COCs were then counted and categorized according to their stages (i.e., GV, pMI, MI, AI/TI, MII, and lysed) (Table 1). Notably, the overall MII rate (i.e., IVM rate), which serves as an indicator of IVM, was significantly higher in type A COCs (11.57 ± 2.48%) compared to type C COCs (1.69 ± 0.89%) (P = 0.02). The MII rate of type A oocytes was also higher than that of type B oocytes (8.33 ± 2.26%), although the difference was not statistically significant (P = 0.266) (Table 1). Thus, type A COCs were used for all subsequent experiments.

### Effects of hormone supplementation in culture medium on the IVM rate of oocytes from natural cycling guinea pig

As shown in Table 1, most type A oocytes, when cultured in the basic medium, stopped meiosis at the MI stage, with only 11.57 ± 2.85% of type A oocytes reaching the MII stage. However, by adding different combinations of hormones, including PMSG, LH, FSH, and hCG, into the basic medium, the ratio of MII oocytes (i.e., IVM rate) increased to varying degrees (Table 2). Notably, the addition of hCG, FSH, and LH resulted in a significant increase in the ratio of MII oocytes, with a maximal rate of 25.10 ± 1.34% (Table 2) ($P < 0.05$). Thus, hCG, FSH, and LH were added to the culture medium for subsequent experiments unless otherwise specified.

### Effects superovulation with different hormones on the IVM rate of oocytes from superovulated guinea pigs

Hormones are commonly used to induce superovulation, which can further affect the IVM of oocytes. In this study, we investigated how different hormones, such as PMSG, FSH and hMG, can affect superovulation as well as the IVM rate of oocytes in guinea pigs. We found that PMSG (25.00 ± 1.73 vs. 20.00 ± 2.51 for PMSG vs. control; $P = 0.359$) and FSH (20.00 ± 2.31 vs. 20.00 ± 2.51 for FSH vs. control; $P = 0.528$) did not increase the number of type A oocytes (Table 3). The MII rates (22.90 ± 2.22% for PSMG, 23.62 ± 1.38% for FSH, vs. 25.10 ± 1.34% for control) ($P = 0.33$ and $0.508$, respectively) of type A oocytes were also not significantly improved ($P = 0.751$) with these two hormones. In contrast, injection of hMG significantly increased the number of type A oocytes (32.33 ± 5.36) compared to injection of FSH (20.00 ± 2.31), PMSG (25.00 ±1.73) or control (20.00 ± 2.51) ($P = 0.029$, $0.154$, and $0.029$, respectively) (Table 4). Similarly, the ratio of MII oocytes from type A oocytes injected with hMG (30.27 ± 1.24%) was significantly higher than those injected with FSH (23.62 ± 1.38%), PMSG (22.90 ± 2.22%) or control (25.10 ± 1.34%) (Table 3) ($P = 0.010$, $0.006$, and $0.036$ respectively). These results indicated that injection of hMG is most effective in inducing superovulation; meanwhile, superovulation induced by hMG can significantly increase the IVM rate of oocytes collected from these superovulated guinea pigs.

**Table 3. Effects of different hormones on superovulation and on the IVM rate of oocytes from these superovulated guinea pigs.**

| Hormone type | Continuous injection time (days) | Number of animals | Number of oocytes | GV (%) | pMI (%) | MI (%) | A/T (%) | MII (%) | Lysed (%) |
|---|---|---|---|---|---|---|---|---|---|
| - | 0 | 10 | 20.00±2.51ab | 3.82±1.90a | 16.78 ±0.47ab | 39.39 ±2.52a | 4.62 ±2.39a | 25.10±1.34a | 10.29±1.16a |
| +PMSG | 1 | 6 | 25.00±1.73b | 6.42±2.15a | 17.81±3.63b | 36.72 ±4.79a | 2.85 ±1.43a | 22.90±2.22a | 13.30 ±0.68ab |
| +FSH | 1 | 6 | 20.00±2.31b | 6.53±0.97a | 16.80±1.08b | 35.56 ±5.29a | 2.08 ±20.8a | 23.62±1.38a | 15.41±1.81b |
| +hMG | 1 | 6 | 32.33±5.36ab | 3.51 ±2.01ab | 13.44 ±3.49ab | 34.2±2.59a | 4.53 ±1.62a | 30.27±1.24b | 14.04±1.37b |
| +hMG | 2 | 6 | 23.00±5.13b | 1.01 ±1.01ab | 11.54 ±0.77ab | 44.92 ±7.40a | 4.16 ±4.16a | 27.84 ±1.82ab | 10.53±1.02b |
| +hMG | 3 | 6 | 39.67±2.60b | 0.00±0.00b | 6.16±2.70a | 52.51 ±3.43a | 4.13 ±0.65a | 31.16±0.86b | 6.04±1.27b |

* Different superscript letters (a, b or ab) in the same column indicate statistical significance between different variables ($P < 0.05$).—indicates no hormone is injected. PMSG, pregnant horse serum gonadotropin; hCG, human chorionic gonadotropin; hMG, human menopausal gonadotropin; GV: germinal vesicle; pMI: pro-metaphase I; MI: metaphase I; AI: anaphase I; TI: telophase I; MII: metaphase II.

**Table 4. Effects of culture duration on the IVM rate of oocytes from superovulated guinea pigs.**

| | Time | Number of animals | Number of oocytes | GV (%) | pMI (%) | MI (%) | A/T (%) | MII (%) | Lysed (%) |
|---|---|---|---|---|---|---|---|---|---|
| Basic medium+hCG+FSH+LH | 24h | 6 | 119 | 0.00±0.00[a] | 6.16±2.70[a] | 52.51±5.95[a] | 4.13±0.65[a] | 31.16±0.86[a] | 6.04±1.27[a] |
| Basic medium+hCG+FSH+LH | 36h | 6 | 123 | 1.11±1.11[a] | 5.08±1.45[a] | 42.00±1.39[b] | 2.14±1.22[a] | 25.40±2.11[ab] | 24.27±0.91[b] |
| Basic medium+hCG+FSH+LH | 48h | 6 | 118 | 2.73±1.77[a] | 3.31±0.48[a] | 41.18±1.74[b] | 1.59±0.80[a] | 18.92±2.56[b] | 32.28±0.53[c] |

\* Different superscript letters (a, b or ab) in the same column indicate statistical significance between different variables ($P < 0.05$). GV: germinal vesicle; pMI: pro-metaphase I; MI: metaphase I; AI: anaphase I; TI: telophase I; MII: metaphase II.

Moreover, the numbers of guinea pig oocytes with hMG treatment (i.e., 1, 2, or 3 days) were higher compared with those without any hMG treatment (32.33 ± 5.36 for 1-day treatment, 23.00 ± 5.13 for 2-day treatment, 39.67 ± 2.60 for 3-day treatment, compared with 20.00 ± 2.51 for control with no treatment) (P = 0.068, 0.621, and 0.010, respectively). The ratios of MII oocytes were also higher compared with those without hMG treatment (30.27 ± 1.24% for 1-day treatment, 27.84 ± 1.82% for 2-day treatment, 31.16 ± 0.8% for 3-day treatment, compared with 25.10 ± 1.34% for control with no treatment) (P = 0.010, 0.077, and 0.005, respectively) (Table 3). Since hMG treatment for 3 days has the highest overall IVM rate, we chose 3 days for supverovulation treatment with hMG for all subsequent experiments.

## Effects of the culturing duration on the IVM rate of oocytes from superovulated guinea pigs

We further investigated how the culture duration can affect the IVM rate of oocytes collected from superovulated guinea pigs. As shown in Table 4, the oocyte lysis rates at 36 h (24.27 ± 0.91%) and 48 h (32.28 ± 0.53%) were significantly higher than that at 24 h (6.04 ± 1.27%) (P < 0.01 for both). Moreover, the ratio of MII oocytes was significantly higher when the oocytes were cultured for 24 h (31.16 ± 0.86%) compared with those cultured for 36 h (25.40 ± 2.11%) or 48 h (18.92 ± 2.56%) (P = 0.086 and 0.005, respectively). Based on these results, we can conclude that 24 h is the optimal culturing time for IVM of guinea pig oocytes, and this duration is used for following experiments unless otherwise specified.

## Effects of different amino acids or in the hibitors on IVM rate of oocytes from superovulated guinea pigs

L-cysteine and cystine play crucial roles in the IVM of oocytes. To investigate their effects on the IVM rate of oocytes from superovulated guinea pigs, we performed a series of tests. When only L-cysteine (200 μM) was added to the culture medium, there was no significant difference in the ratio of MII oocytes (27.74 ± 0.36%) compared to that of the culture medium alone (31.16 ± 0.86%) (P = 0.439). However, when both L-cysteine (200 μM) and cystine (100 μM) were added to the culture medium, the ratio of MII oocytes (48.14 ± 4.96%) significantly increased compared to that of the culture medium alone (31.16 ± 0.86%) (Table 5) (P = 0.006). Inhibitors such as IBMX and ROS can affect the IVM of oocytes and thus we also investigated how they can affect the IVM rate of guinea pig oocytes. When IBMX was added to the maturation medium together with L-cysteine and cystine to a final concentration of 20 μM for 6 h, the ratio of MII oocytes (52.77 ± 6.79%) did not significantly increase compared to the group without IBMX (48.14 ± 4.96%) (P = 0.41) (Table 5). However, addition of ROS together with L-cysteine and cystine for 24 h slightly increased the ratio of MII oocytes compared to the groups without ROS, although not significantly (58.98 ± 5.81% vs. 48.14 ± 4.96%) (P = 0.128) (Table 5). These findings suggest that both L-cysteine and cystine are crucial for IVM of guinea

**Table 5. Effects of different amino acids and inhibitors on the IVM rate of guinea pig oocytes.**

| | Processing time (h) | Number of animals | Number of oocytes | GV (%) | pMI (%) | MI (%) | A/T (%) | MII (%) | Lysed (%) |
|---|---|---|---|---|---|---|---|---|---|
| Basic medium | 0 | 6 | 130 | 2.44 ±0.01 | 21.4 ±0.47a | 35.83 ±4.30ab | 0.78 ±0.77a | 16.05±1.07a | 23.53 ±1.77a |
| Basic medium+hCG+FSH+LH | 0 | 6 | 67 | 0.00 ±0.00 | 6.16 ±0.03bc | 52.51±3.43c | 4.13 ±0.65a | 31.16 ±0.86bc | 6.04 ±1.27bc |
| Basic medium+hCG+FSH+LH +L-Cys | 0 | 6 | 98 | 4.42 ±0.02 | 12.87 ±3.26b | 39.62 ±0.93bc | 3.47 ±2.25a | 27.74 ±26.18ab | 11.88 ±0.58bc |
| Basic medium+hCG+FSH+LH +L-Cys+Cys | 0 | 6 | 81 | 3.10 ±0.19 | 8.23 ±2.85bc | 21.07 ±7.21ad | 5.59 ±3.99a | 48.14 ±4.96de | 13.88 ±5.38bc |
| Basic medium+hCG+FSH+LH +L-Cys+Cys+IBMX | 6 | 6 | 72 | 2.98 ±0.02 | 5.08 ±1.27bc | 21.56 ±6.47ad | 6.84 ±0.63a | 52.77 ±6.79def | 10.77 ±0.33bc |
| Basic medium+hCG+FSH+LH +L-Cys+Cys+IBMX | 8 | 6 | 97 | 2.41 ±0.01 | 7.95 ±1.16bc | 16.73±6.10d | 2.41 ±1.01a | 42.74 ±8.33cd | 9.87 ±2.03bc |
| Basic medium+hCG+FSH+LH +L-Cys+Cys+ROS | 12 | 6 | 54 | 0.00 ±0.00 | 2.98 ±1.65c | 24.28 ±2.53abd | 2.03 ±1.01a | 62.95±2.41f | 7.76 ±1.68bc |
| Basic medium+hCG+FSH+LH +L-Cys+Cys+ROS | 24 | 6 | 74 | 1.73 ±0.09 | 2.59 ±1.48c | 26.91 ±5.02abd | 5.10 ±2.55a | 58.98 ±5.81ef | 4.70±0.49c |

* Different superscript letters (a, b or ab) in the same column indicate statistical significance between different variables (P < 0.05). FBS, fetal bovine serum; hCG, human chorionic gonadotropin; FSH, follicle-stimulating hormone; LH, luteinizing hormone; L- L-Cys, L-cysteine; Cys, cystine; IBMX, diphosphate Esterase inhibitor; ROS, Roscovitine reactive oxygen species inhibitor. GV: germinal vesicle; pMI: pro-metaphase I; MI: metaphase I; AI: anaphase I; TI: telophase I; MII: metaphase II.

pig oocytes, and the addition of ROS could potentially enhance the effects of L-cysteine and cystine.

We further explored how treatment duration of inhibitors such as ROS and IBMX can improve the IVM rate of oocytes. Our results indicate that the duration of inhibitor treatment did not significantly affect the IVM rates. Specifically, there was no significant difference between 6 h (52.77 ± 6.79%) and 8 h (42.74 ± 8.33%) treatment of IBMX regarding the ratio of MII oocytes (P > 0.05) (Table 5). Additionally, there was no significant difference between 12 h (62.95 ± 2.41%) and 24 h (58.97 ± 5.81%) treatment of ROS regarding the ratio of MII oocytes (P = 0.564) (Table 5). However, the rates of MII oocytes and lysed oocytes were higher and lower in the ROS-treated group compared with those of IBMX-treated group, respectively (Table 5), suggesting that ROS is more effective in improving IVM while being less toxic compared with IBMX.

## Effects of treatment dose and duration of hCG on the IVM rate of oocytes from superovulated guinea pigs

Given that oocytes matured *in vivo* have better quality, we explored a combined method of *in vivo* and *in vitro* oocyte maturation. The use of hCG is crucial for *in vivo* maturation of oocytes; thus, we examined how the treatment duration after hCG injection and different doses of hCG could impact the IVM rate of oocytes from superovulated guinea pigs. The ratio of MII oocytes in guinea pigs that received no hCG injection was only 31.16 ± 0.86%; in contrast, the ratios of MII oocytes at 3 h, 6 h and 8 h were significantly higher (66.43 ± 2.57%, 66.43 ± 2.57%, and 69.12 ± 3.80%, respectively with P < 0.001 for all). As the time after hCG injection elapsed, the ratio of MII oocytes decreased to 54.81 ± 3.45% (P < 0.05) at 10 h and 37.69 ± 3.42% (P < 0.05) at 30 h (Table 6). Furthermore, the ratio of MII oocytes in guinea pigs that received 5 IU/kg hCG injection (69.12 ± 3.80%) was significantly higher than those that received 10 IU/kg hCG injection (56.87 ± 3.80%) (P = 0.024) (Table 7). Thus, 5 IU/kg hCG and 8 h were chosen as the condition for *in vivo* oocyte stimulation.

**Table 6. Effects of duration of hCG on the IVM rate of oocytes from superovulated guinea pigs.**

| hCG time (h) | Number of animals | Number of oocytes | GV (%) | pMI (%) | MI (%) | A/T (%) | MII (%) | Lysed (%) |
|---|---|---|---|---|---|---|---|---|
| 0 | 6 | 79 | 0.00±0.00[a] | 6.16±2.70[a] | 52.51±3.43[a] | 4.13±0.65[a] | 31.16±0.86[a] | 6.04±1.27[ab] |
| 3 | 6 | 67 | 1.79±0.89[a] | 1.79±0.89[a] | 25.91±2.78[bcd] | 0.79±0.79[a] | 66.43±2.57[d] | 3.29±2.21[ab] |
| 6 | 6 | 65 | 2.02±2.02[a] | 1.04±1.04[a] | 21.92±3.75[bd] | 4.76±0.93[a] | 65.73±2.10[d] | 4.52±2.29[ab] |
| 8 | 6 | 45 | 0.00±0.00[a] | 5.10±3.04[a] | 16.01±5.99[d] | 6.26±1.67[a] | 69.12±3.80[d] | 3.51±3.50[ab] |
| 10 | 6 | 54 | 1.85±1.85[a] | 1.59±1.58[a] | 31.90±9.31[bc] | 6.03±3.90[a] | 54.81±3.45[c] | 3.81±1.98[ab] |
| 18 | 6 | 38 | 0.00±0.00[a] | 2.78±2.77[a] | 34.52±3.82[bc] | 2.38±.38[a] | 55.16±2.60[c] | 5.16±2.60[ab] |
| 24 | 6 | 29 | 2.78±2.77[a] | 4.76±4.76[a] | 41.51±0.82[ac] | 6.11±3.09[a] | 44.84±2.60[bc] | 0.00±0.00[a] |
| 30 | 6 | 32 | 0.00±0.00[a] | 3.92±3.92[a] | 39.65±3.29[ac] | 9.26±4.89[a] | 37.69±3.42[ab] | 9.48±4.94[b] |

\* Different superscript letters (a, b or ab) in the same column indicate statistical significance between different variables (P < 0.05). hCG time refers to the first time of subcutaneous injection of hCG to removal of ovaries; hCG, human chorionic gonadotropin; GV: germinal vesicle; pMI: pro-metaphase I; MI: metaphase I; AI: anaphase I; TI: telophase I; MII: metaphase II.

## Mature oocytes cultured under the optimized conditions have significantly higher intracellular GSH levels

As shown in Table 8 and Fig 2, the GSH fluorescence intensity was significantly different among groups with different culture media (P < 0.05). Specifically, MII oocytes cultured in the basic medium supplemented with hCG, FSH, LH, L-cysteine, cystine, and ROS exhibited a notably higher GSH fluorescence intensity (80.89 ± 8.63) compared to those cultured in the basic medium (14.40 ± 2.22) (P < 0.05). Moreover, the ratio of MII oocytes significantly increased in the group utilizing basic medium supplemented with hCG, FSH, LH, L-cysteine, cystine, and ROS when compared to that in the group using the basic medium (69.12 ± 3.80% vs. 16.05 ± 1.07%) (P < 0.001), indicating the high quality of mature oocytes induced using our optimized method.

## Mature oocytes cultured under the optimized conditions have normal morphology

As shown in Table 8 and Fig 3, the types of spindles and chromosomes varied in groups utilizing different culture media (P < 0.05 for all). Specifically, basic medium supplemented with hCG, FSH, LH, L-cysteine, cystine, and ROS had a higher ratio (63.19 ± 2.68%) of normal spindle and chromosome distribution aggregates compared to groups utilizing other types of media, indicating that the mature oocytes induced by our optimized method have good morphology and can be used for subsequent studies.

**Table 7. Effects of different doses of hCG on the IVM rate of oocytes from superovulated guinea pigs.**

| hCG injection dose (IU/kg) | Number of animals | Total number of oocytes | GV (%) | pMI (%) | MI (%) | A/T (%) | MII (%) | Lysed (%) |
|---|---|---|---|---|---|---|---|---|
| 0 | 6 | 27 | 0.00±0.00[a] | 6.16±2.70[a] | 52.51±3.43[a] | 4.13±0.65[ab] | 31.16±0.86[a] | 6.04±1.27[a] |
| 5 | 6 | 45 | 0.00±0.00[a] | 5.10±3.04[a] | 16.02±5.99[b] | 6.26±1.67[b] | 69.12±3.80[b] | 3.51±3.50[a] |
| 10 | 6 | 135 | 2.07±1.15[a] | 2.15±1.28[a] | 33.76±3.88[c] | 0.74±0.74[a] | 56.87±3.80[c] | 4.41±1.13[a] |

\* Different superscript letters (a, b or ab) in the same column indicate statistical significance between different variables (P < 0.05). The hCG injection dose refers to the unit of subcutaneous injection of hCG in guinea pigs. The injection volume was in ml. hCG, human chorionic gonadotropin. GV: germinal vesicle; pMI: pro-metaphase I; MI: metaphase I; AI: anaphase I; TI: telophase I; MII: metaphase II.

**Table 8. Effects of different culture media on intracellular levels of GSH and morphology of microtubules in MII oocytes.**

| Culture medium | Total inhibition time (h) | Number of animals | Inhibitor release time | Number of oocytes | MII (%) | Mean intensity of GSH fluorescence | Normal spindle and aggregated chromosomes (%) | Abnormal spindles and diffuse chromosomes (%) |
|---|---|---|---|---|---|---|---|---|
| Basic medium | 0 | 6 | 0 | 110 | 16.05 ±1.07a | 14.4 0±2.22a | 16.67±8.33a | 83.33±8.33a |
| Basic medium+LH +FSH+HCG | 0 | 6 | 0 | 119 | 31.16 ±0.86b | 15.89±5.01a | 42.26±2.04b | 57.74±2.04b |
| Basic medium+LH +FSH+HCG +L-Cys+Cys | 0 | 6 | 0 | 97 | 48.14 ±4.96c | 43.71±6.21b | 48.33±4.68b | 51.67±4.68b |
| Basic medium+LH +FSH+HCG +L-Cys+Cys+ROS | 12 | 6 | 6 | 84 | 69.12 ±3.80d | 80.89±8.63c | 40.67±1.62b | 59.33±1.62b |
| Basic medium+LH +FSH+HCG +L-Cys+Cys+ROS | 24 | 6 | 12 | 104 | 58.97 ±5.81c | 75.53±9.28c | 63.19±2.68c | 36.81±2.68c |

* Different superscript letters (a, b or ab) in the same column indicate statistical significance be man chorionic gonadotropin; FSH, follicle-stimulating hormone; LH, luteinizing hormone; L-Cys, L-cysteine; Cys, cystine; ROS, roscovitine reactive oxygen species inhibitor.* Different superscript letters (a, b or ab) in the same column indicate statistical significance between different variables ($P < 0.05$). FBS, fetal bovine serum; hCG, human chorionic gonadotropin; FSH, follicle-stimulating hormone; LH, luteinizing hormone; L-Cys, L-cysteine; Cys, cystine; ROS, roscovitine reactive oxygen species inhibitor. GV: germinal vesicle; pMI: pro-metaphase I; MI: metaphase I; AI: anaphase I; TI: telophase I; MII: metaphase II.

## Discussion

The immature oocyte is typically enclosed by cumulus cells with varying layers and shapes, collectively referred to as COCs. The cumulus cells are crucial for oocyte maturation, with their number and morphology serving as potential indicators of oocyte maturation. Previous studies have suggested that the thicker the cumulus cell layer, the greater the likelihood of oocyte maturation *in vivo*. In this study, we categorized COCs into three types based on cumulus cell layers (A, B, and C types). As per previous findings, we observed that type A COCs with the maximum number of cumulus cell layers had the highest percentage of mature oocytes, while type C COCs, with almost no cumulus cells, had a very low percentage of mature oocytes (i.e., barely detectable MII). These observations highlight the critical role of cumulus cells in oocyte maturation, and suggest that type A COCs, which have the maximal cumulus cell layers, should be used in *in vitro* maturation experiments with guinea pig oocytes.

The synthesis of glutathione (GSH) is crucial for oocyte maturation, with *in vivo* mature oocytes having a significantly higher GSH concentration than *in vitro* mature oocytes [22]. Previous studies have shown that the addition of L-cysteine during the IVM of oocytes from various species (e.g., cattle, pigs, and buffalo) can promote the synthesis of intracellular GSH [23–27], thereby promoting oocyte maturation. Cystine has been widely used in mammalian oocyte IVM experiments to increase cytoplasmic GSH content, which can promote oocyte maturation [22]. Consistent with these previous studies, we found that the addition of L-cysteine and cystine significantly enhanced the proportion of oocytes reaching the M II stage, indicating their capacity to promote IVM of guinea pig oocytes.

Hormones play a critical role in the maturation and ovulation of oocytes *in vivo* [28]. FSH or LH is commonly used for IVM [29, 30], with FSH binding to FSH receptors on the surface of follicular cells, inducing estradiol and LH secretion. Finally, under the action of LH, the oocyte begins to resume meiosis, the nuclear membrane ruptures and chromosome condensation occurs. The oocyte will not be expelled from the ovary until the first meiosis is completed. During the process of oocyte maturation, appropriate concentrations of FSH and LH in the *in*

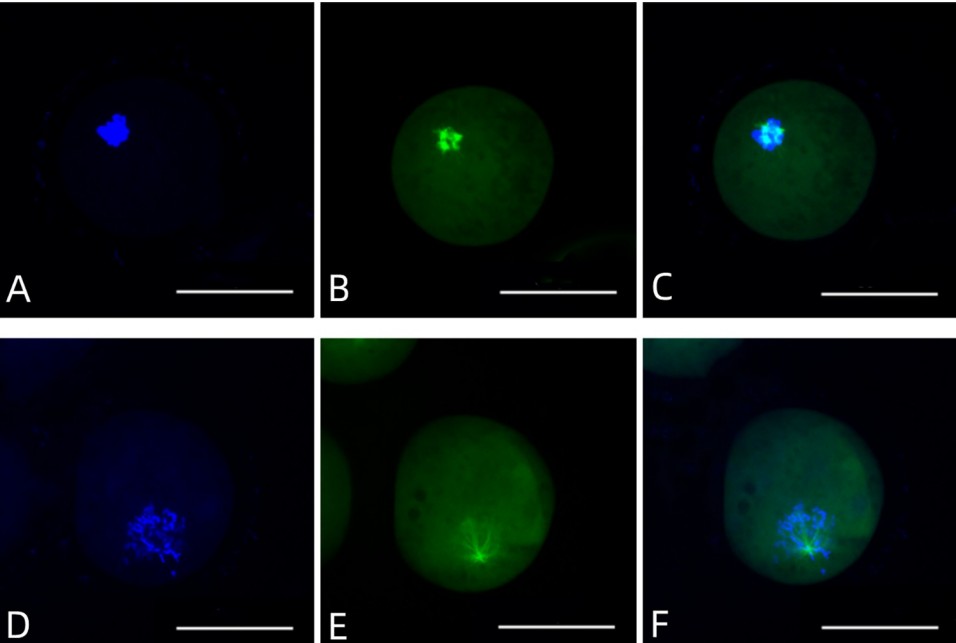

**Fig 2. Types of oocytes from guinea pigs and their meiotic progression during IVM.** The arrow indicates the first polar body of the matured oocytes, indicating the oocytes have entered the MII phase. (A) Matured oocytes stained with Cell Truck Bule CMF2HC under the microscope after treatment with ROS for 12 hours. (B) Immature oocytes stained with Cell Truck Bule CMF2HC under the microscope after treatment with ROS for 12 hours. (C) After the removal of IMBX, the fluorescence intensity of matured oocytes treated with IBMX was significantly stronger than that treated with ROS. (D) The fluorescence intensity of GSH in MII oocytes cultured in the maturation medium was relatively weaker, and the first polar body was not obvious as well, compared with that cultured in the maturation medium with ROS. (E) The fluorescence intensity of GSH in MII oocytes cultured in MTA medium supplemented with L-cysteine and cystine was stronger than that of GSH in MII oocytes cultured in basic medium. (F) The fluorescence intensity of GSH cultured in the basic medium was much lower than other groups.

*vitro* maturation culture medium play a significant role. In the present study, similar to what was observed in bovine oocytes [29], FSH and LH promoted the maturation of guinea pig oocytes during IVM; the promoting effect is more prominent in type A oocytes than in type B oocytes. It is worth mentioning that in similar experiments conducted by other researchers, the MII rate of type A oocytes cultured in basic medium+HCG+LH+FSH+Cys+LIF (leukemia inhibitory factor) was 61.8%, while the MII rate of those cultured in basic medium+HCG+LH +FSH+Cys+L-Cys+ROS in our study was 69.12% [31]. In one study, the increase in MII rates induced by M199+10%FSB, M199+10%FBS+LH+FSH, and M199+10%FBS+LH+FSH+Cys were 17.5%, 41.4% and 39.5%, respectively, similar to what we observed, showing a consistent trend in the increase of MII rates [31]. However, when cultured with the same medium, the MII rate in their study [31] was slightly higher than that in ours, which may be due to differences in experimental animals used, environmental conditions, and methods used for superovulation.

ROS is a potent CDK inhibitor that can concentration-dependently inhibit the recovery of meiosis in oocytes such as bovine [32], pig [33], and goat [34]; the oocytes can resume meiosis after the inhibition by ROS is released. The duration of ROS inhibition can also affect IVM efficiency. For instance, in pig oocytes, 50 μM of ROS was used to inhibit pig oocytes for 30 h, leading to 86.2% of the oocytes accumulating in the GV stage; while without ROS inhibition, 85% of the oocytes can reach the MII stage [35]. However, under the same conditions, it was also found that 25 μM of ROS could inhibit the recovery of meiosis in pig oocytes, but the

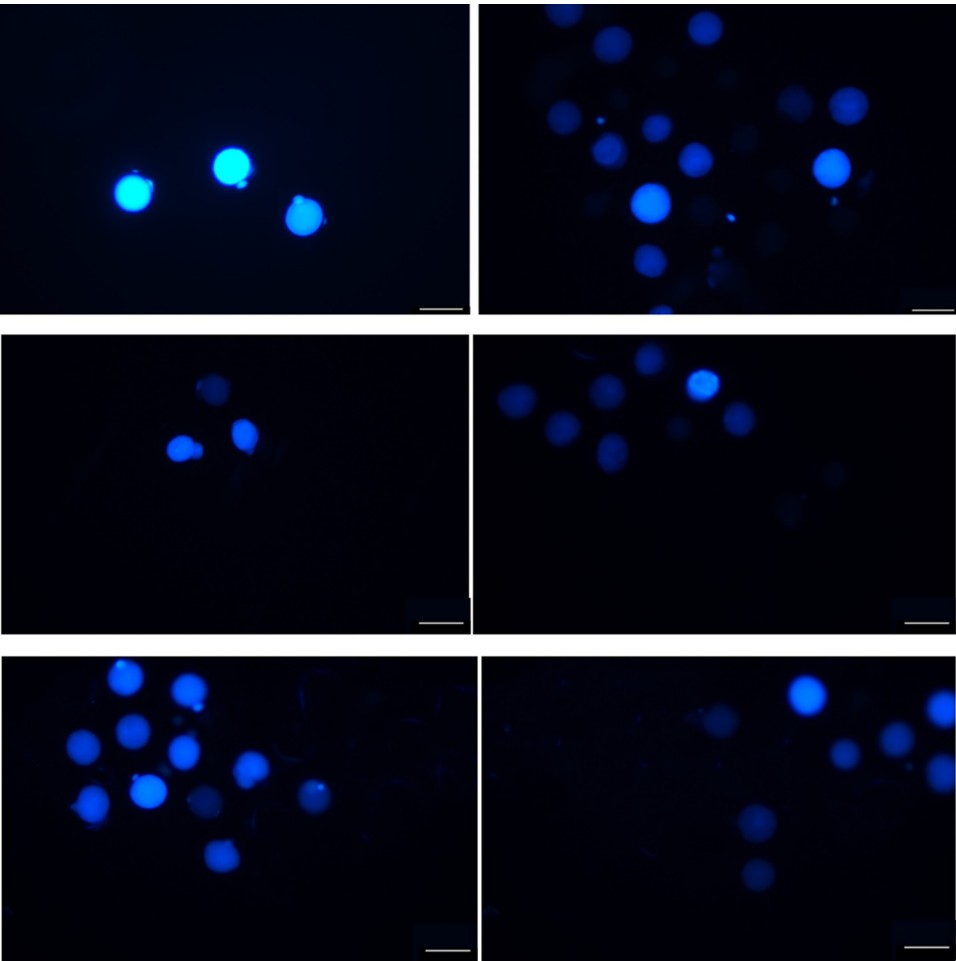

**Fig 3. The microtubule morphology of MII oocytes from guinea pigs was divided into two types.** For MII oocytes with normal spindles and aggregated chromosomes (A, B and C in the top panel), the chromosomes were distributed and aggregated on the equatorial plate, while for MII oocytes with abnormal spindles and dispersed chromosomes (D, E and F in the bottom panel), the chromosomes could not be neatly distributed on the equatorial plate and are in a diffusion state. A and D are for Hoechst 33342 staining, B and E are for β-tubulin staining, and C and F are merged for A and B, and D and E, respectively. Bar in 50 μm.

fertilization rate and blastocyst rate decreased after this inhibition [36]. In 2005, it was also suggested that long-term inhibition with high concentrations of ROS would damage the developmental ability of oocytes [34]. In this study, we found that ROS inhibition time needs to be controlled within a suitable range; otherwise, the IVM rate of oocytes will be affected. In addition, we found that ROS is more effective than IBMX in promoting IVM of guinea pig oocytes while having fewer side effects.

## Conclusion

In this study, we reported the successful establishment of an IVM method for guinea pig oocytes, which can yield an IVM rate of 69%. We showed that superovulation with hMG, supplementation of hormones such as hCG, FSH and LH, amino acids such as L-Cys and Cys, and inhibitors such as ROS and IMBX in the culture medium can increase the IVM rate. Moreover, *in vivo* hCG stimulation can further increase the IVM rate. Thus, our research provides a basis for further study of *in vitro* fertilization (IVF) and embryo engineering, among

others, using guinea pigs as model animals. Our study might also make guinea pigs more popular as an animal model for the study of reproduction diseases and other diseases.

## Supporting information

**S1 Dataset.**
(XLSX)

## Author Contributions

**Conceptualization:** Xinghua Xu, Mingjiu Luo, Hongshu Sui.

**Formal analysis:** Minhua Yao, Zhaoqing Gong, Xinlei Shi, Xiaocui Liu.

**Funding acquisition:** Xinghua Xu, Mingjiu Luo, Hongshu Sui.

**Investigation:** Minhua Yao, Zhaoqing Gong, Weizhen Xu.

**Methodology:** Minhua Yao, Zhaoqing Gong, Weizhen Xu.

**Project administration:** Hongshu Sui.

**Resources:** Yangyang Tang, Siyu Xuan, Yanping Su.

**Software:** Yangyang Tang, Siyu Xuan, Yanping Su.

**Supervision:** Hongshu Sui.

**Writing – original draft:** Minhua Yao, Xinghua Xu, Hongshu Sui.

**Writing – review & editing:** Hongshu Sui.

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
