## [Decision Letter · Decision Letter 0]

10 Jan 2023

PONE-D-22-31747Establishment and optimization of an in vitro  guinea pig oocyte maturation system using L-cysteine, cystine, and roscovitinePLOS ONE

Dear Dr. Sui,

Thank you for submitting your manuscript to PLOS ONE. After careful consideration, we feel that it has merit but does not fully meet PLOS ONE’s publication criteria as it currently stands. Therefore, we invite you to submit a revised version of the manuscript that addresses the points raised during the review process.

We look forward to receiving your revised manuscript.

Kind regards,

Wei Cui, Ph.D.

Academic Editor

PLOS ONE

Journal Requirements:

Additional Editor Comments:

Please refer to the Reviewers' comments for details.

Reviewers' comments:

Reviewer's Responses to Questions

**Comments to the Author**

1. Is the manuscript technically sound, and do the data support the conclusions?

Reviewer #1: No

Reviewer #2: Yes

2. Has the statistical analysis been performed appropriately and rigorously? 

Reviewer #1: I Don't Know

Reviewer #2: Yes

3. Have the authors made all data underlying the findings in their manuscript fully available?

Reviewer #1: No

Reviewer #2: Yes

4. Is the manuscript presented in an intelligible fashion and written in standard English?

Reviewer #1: No

Reviewer #2: No

5. Review Comments to the Author

Reviewer #1: There are many ambiguities in this work, which raises doubts as to the reliability of the results. Therefore, I refuse to publish this manuscript.

The Materials and Methods as well as Results sections must be completely rewritten. The description of the whole experiment is very chaotic, incoherent, even incomprehensible to the reader. A lot of important information is missing:

Detailed notes:

- please provide the approval number of the Animal Research Ethics Committee

-please provide the number of animals used for the experiment and their weight

- please, clearly and legibly describe this experiment

Line 100: Authors should add the word "respectively" for the given doses of hMG and hCG

Line 109: What dose of anesthetic was used to euthanize the animals?

Line 111: How many follicles and what types of follicles were oocytes isolated? (diameter?) How many animals were used in this experiment?

Lines 118-125: Authors should clearly define what the composition of the basic and maturation medium is. There are some discrepancies in the description of the methodology and results (data in the tables), which causes the reader to be lost and confused.

lines 127-134. Authors should briefly characterize the tested inhibitors. Also, the description is unclear.

Results

Line 178-18: This description should be in the Materials and Methods section

Line 178: It is said: These follicles are divided into 3 types ... it should rather be that COCs isolated from the follicles are divided into ...

Table 1 - presents data on animals without superovulation - is this a control group? - In the Materials and Methods chapter, the authors do not mention it at all

Lines 190-191. Delete the sentence starting with: “Type A oocytes have more than (…. )” This information is given in the description of Figure 1

Table 2 - What is the composition of the basic medium? - only FBS or FBS+PMSG - unclear description in Materials and Methods,

Table 3: Where is the description of this experiment in the Materials and Methods section?

Table 5: these 24, 36, 48 hour incubation times should be clearly indicated in the Materials and Methods section. The lack of such information causes consternation in the reader.

Tables 6-7: They are completely incomprehensible and illogical, the data presented in Table 6 are duplicated in Table 7.

Lines: 303-304: description inadequate compared to the information in the table.

Tables 8 and 9 - data are very questionable. Data for MAT+L-Cys+Cys culture medium (Table 8) are almost identical to data for MAT+IBMX+L-Cys+Cys in Table 9. In addition, data for MII (%) in MAT+ L-Cys+Cys and MAT+L-Cys+Cys+IBMX (Table 8) have been replaced by MAT+L-Cys+Cys+IBMX (6 and 8 hours of incubation) in Table 9. It is generally unclear why the authors included Table 8 in this article as the same is shown in Table 9 (extended by incubation times).

The same problem goes for Tables 7 and 8 - Is the culture medium M199+10%FBS+hCG equivalent to M199+10%FBS??? or is similarly MAT+L-Cys+Cy (Table 8) the same as M199+10%FBS+hCG+LH+FSH+L-Cys+Cys (Table 9)?? because both tables show the same data. Why did the authors change the abbreviations of the media descriptions? This confuses the reader (apart from the reliability of the data given in the tables). Authors should standardize the description and use the same abbreviations throughout the manuscript.

Line 351-355: It is said- In maturation medium supplemented with L-cysteine, cystine and ROS, the ratio of MII oocytes from guinea pigs with no hCG injection was only 31.16 ±

0.86% - The same data are given in Table 7 for the culture medium composed of M199+10%+FBS+hCG+LH+FSH. It is completely incomprehensible

The manuscript requires reanalysis and re-presentation of the results obtained. In addition, a linguistic correction is required, and in many places in the manuscript the authors did not report statistical significance (p<00000).

Reviewer #2: Manuscript Number: PONE-D-22-31747

Title: Establishment and optimization of an in vitro guinea pig oocyte maturation system using L-cysteine, cystine, and roscovitine

Authors: Minhua Yao, Zhaoqing Gong, Weizhen Xu, Xinlei Shi, Xiaocui Liu , Yangyang Tang, Siyu Xuan, Yanping Su, Xinghua Xu1*, Mingjiu Luo, Hongshu Sui

General comments:

Authors have performed a series of experiments to demonstrate that basic medium supplemented with L-cysteine, cystine, and ROS can increase the IVM rate of guinea pig oocytes, and that injecting hCG in guinea pigs can further increase the IVM rate by 69% when the oocytes used for following IVM are collected 3-8 h after injection. The reviewer believes that the MS is interesting to be published.

Some other comments:

1. The similar work has been published by others Wang et al., Am J Transl Res 2019;11(12):7479-7491 www.ajtr.org /ISSN:1943-8141/AJTR0103814, please refer this in MS. The reviewer suggests that the comparison of your data with that published is necessary in the discussion.

2. Shrink the Table into 6 tables, while others were either stated in the text or in figure. For example, Tables 1-11 are combined into important 6 tables; Table 12, 13 are combined into one figure.

3. The English style needs to be edited by professional editing company.

6. PLOS authors have the option to publish the peer review history of their article (what does this mean?). If published, this will include your full peer review and any attached files.

Reviewer #1: No

Reviewer #2: No

---

## [Author Response · Author response to Decision Letter 0]

21 Mar 2023

Responses to Reviewers

Reviewer #1

1.There are many ambiguities in this work, which raises doubts as to the reliability of the results.

Reply: We sincerely thank the reviewer’s comments and suggestions, which are indeed helpful for us to improve our writing and data presentation. We agree that some description and data presentation in the original manuscript is not clear enough; thus, we have thoroughly checked our manuscript and reorganized the whole manuscript in writing and data presentation. In reflection of these changes, we have modified our manuscript title from “Establishment and optimization of an in vitro guinea pig oocyte maturation system using L-cysteine, cystine, and roscovitine” to “Establishment and optimization of an in vitro guinea pig oocyte maturation system”.

2.The Materials and Methods as well as Results sections must be completely rewritten. The description of the whole experiment is very chaotic, incoherent, even incomprehensible to the reader. 

Reply: Thanks for your suggestion. We have now completely rewritten the Materials and Methods section, as well as the Results section. For the Materials and Methods section, we have ensured that all media, chemicals, reagents, and experimental procedures are described with details. For the logic of the manuscript, we have made significant changes. Basically, we treated all the conditions, such as COC types, superovulation by different hormones, culture supplementation (i.e., amino acids, hormones and inhibitors), and in vivo stimulation with hCG, as factors and explore how they affect IVM rate in guinea pig oocytes.

3. Detailed notes: please provide the approval number of the Animal Research Ethics Committee; please provide the number of animals used for the experiment and their weight; please, clearly and legibly describe this experiment.

Reply: The number of animals used has been added in each table (i.e., Tables 1-8). The average weight of the animals was ~80 g. This information has now been added to line 115. Each guinea pig was weighed when performing the injection but not listed as the weight of each mouse is comparable (i.e., ~80 g). The approval number of the Animal Research Ethics Committee has now been added (line 112). We have checked the whole manuscript carefully and have almost completely rewritten some sections (especially the Material and Methods, and Results sections) to make sure all the experiments are clearly described and all the data are presented with enough details.

4. Line 100: Authors should add the word "respectively" for the given doses of hMG and hCG

Reply: Thanks for your suggestion. This has now been modified. In addition, the whole manuscript has now been further polished and copyedited by TopEdit, a professional copyediting service provider.

5.Line 109: What dose of anesthetic was used to euthanize the animals?

Reply: For anesthetization, 0.2 ml of 5% chloral hydrate was used per 10 g of body weight. This information has now been added (line 131).

6.Line 111: How many follicles and what types of follicles were oocytes isolated? (diameter?) How many animals were used in this experiment?

Reply: The average number of follicles, including types A, B, and C, before superovulation was ~30, and after superovulation was varied (in average ~50). Except for data in Table 1, where all 3 types of follicles were collected, for other experiments, only type A follicles were dissected for experiments. The number of oocytes was listed in each table (i.e., in Table 1, refers to types A, B, and C; in Table 2-8, refers to only type A). 

7. Lines 118-125: Authors should clearly define what the composition of the basic and maturation medium is. There are some discrepancies in the description of the methodology and results (data in the tables), which causes the reader to be lost and confused.

Reply: The basic medium contains M199 + 10% FBS, and the maturation medium can be different in different situations. Thus, in the revised version, we did not use the term “maturation medium” or “MAT”; instead, we used basic medium plus the exact components to refer to the exact medium used.

6.lines 127-134. Authors should briefly characterize the tested inhibitors. Also, the description is unclear.

Reply: More detailed characteristics of these two inhibitors were added to the introduction part (line 75-79).

7.Line 178-180: This description should be in the Materials and Methods section

Reply: Thank you for your suggestion. This is described in the “Collection of cumus oocyte complexes (COCs)” part of the Materials and Methods section. Moreover, we have thoroughly checked the manuscript and ensured all details for experimental procedures are described in the Materials and Methods section.

8.Line 178: It is said: These follicles are divided into 3 types ... it should rather be that COCs isolated from the follicles are divided into ...

Reply: Thank you for pointing out this error. This has now been modified (line xxx). Moreover, we have checked throughout the manuscript to ensure that such descriptions are accurate.

9.Table 1 - presents data on animals without superovulation - is this a control group? - In the Materials and Methods chapter, the authors do not mention it at all

Reply: We apologize for the confusion. Actually, for data in Table 1 and Table 2, oocytes from natural cycling guinea pigs were used, while for the rest of the Tables, oocytes from superovulated oocytes were used. Data in Table 1 was used to compare how different COC types can affect the IVM rate of oocytes. To make this point clear, we have now stated the origin of the oocytes (e.g., lines 227-228). Regarding the COC collection, we have also now added the description “natural cycling or superovulated guinea pigs” (lines 130-131). Moreover, we have reorganized the logics in the whole manuscript to avoid such confusion.

10.Lines 190-191. Delete the sentence starting with: “Type A oocytes have more than (…. )” This information is given in the description of Figure 1

Reply: Thanks for your suggestion. This has now been deleted.

11.Table 2 - What is the composition of the basic medium? - only FBS or FBS+PMSG - unclear description in Materials and Methods

Reply: We agree the description of the medium composition is not clear. Thus, we have now defined the basic medium as M199 + 10% FBS, and all the other media as basic medium + exact components. In addition, we have now described what types of medium were used (lines 137-177).

12 Table 3: Where is the description of this experiment in the Materials and Methods section?

Reply: We regret this is not clearly stated in the original manuscript. This has now been added in the “Superovulation in guinea pigs” part of the Materials and Methods section (lines 119-127).

13.Table 5: these 24, 36, 48 hour incubation times should be clearly indicated in the Materials and Methods section. The lack of such information causes consternation in the reader.

Reply: Thank the reviewer for pointing out this. We have now added the detailed information for this experiment (line xxx153). Moreover, we have added detailed information for other experiments where necessary.

14.Tables 6-7: They are completely incomprehensible and illogical, the data presented in Table 6 are duplicated in Table 7.

Reply: Thanks for pointing out this error. We have now merged the original Table 6 and Table 7 to make it more sound and easy to understand.

15.Lines: 303-304: description inadequate compared to the information in the table.

Reply: We have now added detailed information regarding the conditions used for data in each table (lines 137-177).

16.Tables 8 and 9 - data are very questionable. Data for MAT+L-Cys+Cys culture medium (Table 8) are almost identical to data for MAT+IBMX+L-Cys+Cys in Table 9. In addition, data for MII (%) in MAT+ L-Cys+Cys and MAT+L-Cys+Cys+IBMX (Table 8) have been replaced by MAT+L-Cys+Cys+IBMX (6 and 8 hours of incubation) in Table 9. It is generally unclear why the authors included Table 8 in this article as the same is shown in Table 9 (extended by incubation times).

Reply: Thanks for pointing out this error. We originally arranged in this way thinking it might be easier for comparison. We realized this does not make a lot of sense as the data are duplicated in this way. We have now merged the data in Table 8 and Table 9. Moreover, we have redefined the basic medium and the exact components in each medium used.

17.The same problem goes for Tables 7 and 8 - Is the culture medium M199+10%FBS+hCG equivalent to M199+10%FBS??? or is similarly MAT+L-Cys+Cy (Table 8) the same as M199+10%FBS+hCG+LH+FSH+L-Cys+Cys (Table 9)?? because both tables show the same data. Why did the authors change the abbreviations of the media descriptions? This confuses the reader (apart from the reliability of the data given in the tables). Authors should standardize the description and use the same abbreviations throughout the manuscript.

Reply: Thanks for pointing out this error. This has now been fixed. We have now standardized the description for the culture medium and abbreviations.

18. The manuscript requires reanalysis and re-presentation of the results obtained. In addition, a linguistic correction is required, and in many places in the manuscript the authors did not report statistical significance (p<00000).

Reply: Thanks for the reviewer’s suggestion. The whole manuscript has now been reorganized and the data have been presented in a different way based on your suggestion. We have added all exact p values or their ranges.

Reviewer-2.

1.Authors have performed a series of experiments to demonstrate that basic medium supplemented with L-cysteine, cystine, and ROS can increase the IVM rate of guinea pig oocytes, and that injecting hCG in guinea pigs can further increase the IVM rate by 69% when the oocytes used for following IVM are collected 3-8 h after injection. The reviewer believes that the MS is interesting to be published.

Reply: We sincerely thank the reviewer for the comments and suggestions.

2.The similar work has been published by others Wang et al., Am J Transl Res 2019;11(12):7479-7491 www.ajtr.org /ISSN:1943-8141/AJTR0103814, please refer this in MS. The reviewer suggests that the comparison of your data with that published is necessary in the discussion.

Reply: We regret that we have omitted this important literature. We have now added the detailed comparison in the discussion part (lines 488-497).

3.Shrink the Table into 6 tables, while others were either stated in the text or in figure. For example, Tables 1-11 are combined into important 6 tables; Table 12, 13 are combined into one figure.

Reply: Thanks for your suggestion. We have now merged tables where data are duplicated.

4.The English style needs to be edited by professional editing company.

Reply: The manuscript has now been edited by a professional editing company, and many parts of the manuscript have been rewritten to ensure the experimental procedures and data are presented in a clear and logic way.

---

## [Decision Letter · Decision Letter 1]

14 Apr 2023

Establishment and optimization of an in vitro  guinea pig oocyte maturation system

PONE-D-22-31747R1

Dear Dr. Sui,

We’re pleased to inform you that your manuscript has been judged scientifically suitable for publication and will be formally accepted for publication once it meets all outstanding technical requirements.

Kind regards,

Wei Cui, Ph.D.

Academic Editor

PLOS ONE

Additional Editor Comments (optional):

Questions and concerns have been well addressed.

Reviewers' comments:

Reviewer's Responses to Questions

**Comments to the Author**

1. If the authors have adequately addressed your comments raised in a previous round of review and you feel that this manuscript is now acceptable for publication, you may indicate that here to bypass the “Comments to the Author” section, enter your conflict of interest statement in the “Confidential to Editor” section, and submit your "Accept" recommendation.

Reviewer #2: All comments have been addressed

2. Is the manuscript technically sound, and do the data support the conclusions?

Reviewer #2: Yes

3. Has the statistical analysis been performed appropriately and rigorously? 

Reviewer #2: Yes

4. Have the authors made all data underlying the findings in their manuscript fully available?

Reviewer #2: Yes

5. Is the manuscript presented in an intelligible fashion and written in standard English?

Reviewer #2: Yes

6. Review Comments to the Author

Reviewer #2: All comments are addressed. The MS revision is ready for publishing. The reviewer accept this MS for publication in this journal.

7. PLOS authors have the option to publish the peer review history of their article (what does this mean?). If published, this will include your full peer review and any attached files.

Reviewer #2: No

---

## [Editor Report · Acceptance letter]

20 Apr 2023

PONE-D-22-31747R1 

Establishment and optimization of an *in vitro* guinea pig oocyte maturation system 

Dear Dr. Sui:

I'm pleased to inform you that your manuscript has been deemed suitable for publication in PLOS ONE. Congratulations! Your manuscript is now with our production department. 

Kind regards, 

on behalf of

Prof. Wei Cui 

Academic Editor

PLOS ONE